# Catechins and Selenium Species—How They React with Each Other

**DOI:** 10.3390/molecules28155897

**Published:** 2023-08-05

**Authors:** Aleksandra Sentkowska, Krystyna Pyrzynska

**Affiliations:** 1Heavy Ion Laboratory, University of Warsaw, Pasteur Str. 5A, 02-093 Warsaw, Poland; sentkowska@slcj.uw.edu.pl; 2Faculty of Chemistry, University of Warsaw, Pasteur Str. 1, 02-093 Warsaw, Poland

**Keywords:** catechins, selenium species, stability, interactions

## Abstract

The combination of selenium and tea infusion, both with antioxidant properties, has potentially complementary mechanisms of action. Se-enriched tea has been considered as a possible Se supplement and a functional beverage to reduce the health risk of Se deficiency. This work investigated the interactions between plant catechins present in tea infusions and selenium species based on changes in the concentration of both reagents, their stability in aqueous solutions, and the possibilities of selenonanoparticles (SeNPs) formation. Selenium species exhibited instability both alone in their standard solutions and in the presence of studied catechins; selenocystine appeared as the most unstable. The recorded UV–Vis absorption spectra indicated the formation of SeNPs in the binary mixtures of catechins and selenite. SeNPs have also formed with diameters smaller than 100 nm when selenite and selenomethionine were added to tea infusions. This is an advantage from the point of view of potential medical applications.

## 1. Introduction

Catechins are members of the flavonoids family of phenolic compounds. They belong to the subgroup of flavanol (also known as flavan-3-ols) and are distributed in a variety of foods and herbs. Green tea is one of the most well-known and richest sources of catechins, mainly epigallocatechin gallate (EGCG) [1,2]. These compounds are thermally sensitive and undergo epimerization to catechin or epicatechin and later degradation to protocatechuic and phloroglucinol acids; this process is influenced by food components and pH values [3]. The beneficial effects of catechins on human health are mainly due to their antioxidant activity and are used in the prevention and treatment of various diseases [4]. Catechins exert their antioxidant properties through several mechanisms, including direct scavenging of reactive oxygen species (ROS), chemical reducing activity, activation of antioxidant enzymes, and inhibition of oxidases [5,6]. The mechanisms of tea catechins-mediated cancer prevention were recently summarized by Cheng et al. [7]. The stability of catechins is very important for their absorption into the human body and the effectiveness of their antioxidant properties. However, the low plasma bioavailability of catechins is considered the major reason for the controversial effects between in vitro and in vivo studies [8]. Due to their useful actions, catechins are increasingly used in medical, pharmaceutical, and cosmetic products [9,10]. Catechin can reduce the absorption of cholesterol in the intestine by forming insoluble precipitation [10]. The modification of Se-doped hydroxyapatite with catechins from green tea extract was proposed for potential application in osteosarcoma therapy [11]. The presence of these compounds prevents the adverse and toxic effects of high concentrations of selenium on normal cells for bone cancer therapy. Green tea extracts are also used in the food industry, as catechins have a high potency for the prevention of lipid oxidation in meat and meat products [12].

Selenium is an essential trace element that is required for several biological functions [13,14]. It is the main constituent of antioxidant enzymes such as thioredoxin reductase, iodothyronine deiodinases, and glutathione peroxidase, which are crucial in scavenging reactive oxygen species [15]. The deficiency of selenium has been linked to a range of serious conditions such as cardiovascular and inflammatory diseases, diabetes, and many others [16,17,18]. However, at high doses, Se stimulates the formation of ROS and exhibits pro-antioxidant activity [19]. To enrich tea plants with selenium, particularly in regions with low content in soils, different biofortification strategies can be used by applying Se-containing fertilizers or by spraying selenium salts onto plant leaves [20,21,22].

The biological activities of selenium are determined not only by the intake dose but also by its chemical form. Compared with inorganic selenium species (selenite and selenate), its organic compounds, such as selenoaminoacids, are less toxic and have greater bioavailability [23]. The newest investigated Se form, selenium nanoparticles (SeNPs), has the advantages of high biological activity and low toxicity. They can be directly absorbed by the human body, having broad prospects in medical applications [24,25]. SeNPs can be synthesized by chemical reducing agents, phytoconstituents of plant extracts, and microorganisms [26,27,28]. The secondary plant metabolites act as selenium reducers but also prevent the aggregation of NPs and promote the production of smaller particles. Green synthesis of SeNPs using aqueous tea extracts has also been used [29,30,31]. The obtained selenium nanoparticles showed a greater ability to neutralize hydroxyl radicals compared to tea extracts [29].

Selenium and green tea, both play an important role in antioxidant defense systems, are particularly interesting as a combination because they have potentially complementary mechanisms of action [32,33,34]. Se-enriched tea, processed from plants grown in seleniferous areas in China, and obtained by using selenium fertilizers has become increasingly popular in recent years due to its more pronounced health-promoting properties [35,36,37]. Moreover, Se-enriched tea has been considered as a possible Se supplement and a functional beverage to reduce the health risk of Se deficiency [38,39]. The interactions between tea polyphenols and selenium species have been studied previously but in the context of the antioxidant properties of their mixtures [40]. The results of the experiments for selenium species and tea infusion binary mixtures were compared with those obtained by adding up the effects of both individual components analyzed separately. The antioxidant activity of green tea infusion with selenium compound evaluated by DPPH assay decreased in the order: MeSeCys > Se(VI) > SeMet~Se(IV). However, in those studies, the formation of selenium nanoparticles was not taken into account.

This work aimed to investigate how different chemical forms of selenium, which are naturally present in plant materials, react with popular catechins based on their concentration changes and possibilities of SeNPs formation. High-performance liquid chromatography in HILIC mode was used to study the changes in the content of catechin, epicatechin, and epigallocatechin gallate (EGCG) in the presence of inorganic and organic selenium species. This issue has not been described before, and the presented results may be valuable in the design of both functional foods and dietary supplements.

## 2. Results and Discussion

### 2.1. Catechins and Selenium Species in the Model Solutions

The mixtures of the main catechins (catechin, epicatechin, and EGCG) at a concentration of 5.00 mg L^−1^ and selenium standard compounds such as Se(IV), Se(VI), selenomethionine (SeMet), selenomethionine oxide (SeMetO), selenocystine (SeCys_2_), and metyloselenocysteine (MeSeCys) at the same concentration were prepared in water. The concentrations of the mixture components were determined by HPLC after every 4 h. Each sample was analyzed in triplicate, and the results expressed as a mean ± standard deviation are presented in Figure 1, Figure 2 and Figure 3. Between subsequent analyses, the studied binary mixtures were stored at room temperature and protected from light.

The obtained results showed that SeMetO was included in the study due to the reported oxidation of selenomethionine [41,42,43,44]. The SeMet oxidation process, which takes place during sample extraction or storage, can be observed by the appearance of an additional chromatographic signal identified as SeMetO, with a simultaneous decrease in the intensity of the SeMet peak. Amako et al. reported that SeMetO had started to appear as soon as selenized yeast was dried and tableted [42]. It was confirmed that selenomethionine was the only source of SeMetO in the certified reference materials of a different matrix using HPLC ICP-MS and the addition of spike the solution before extraction [44].

Generally, catechin does not react with Se(VI) species as their concentrations in the binary mixtures changed only a little, while in the solution containing catechin and Se(IV), a gradual decrease in both components can be observed, more visible for phenolic compound (Figure 1). Catechin, due to its reductive properties, transforms Se oxyanion to elemental selenium [45]. Catechin concentrations in the presence of SeMet decreased slightly and after 24 h were 87.8% of its initial value. In parallel, a more significant decrease in selenomethionine content was determined, and traces of SeMetO appeared. The concentration of that selenium species with catechin was also decreasing at a prolonged time of their mixing, however, much faster than in the case of selenomethionine. An increasing concentration of SeMet was detected in this solution, which may indicate a reduction in selenomethionine oxide to selenomethionine. However, the sums of these two species were lower in comparison to the initial level of SeMet. Probably other selenium species are formed, and the decomposition products change depending on the degree of its advancement [46,47]. According to Gammelgaard et al., the first stage of the SeMet oxidation process is Se-dihydroxy-selenomethionine, followed by cyclization and the formation of methylseleninic acid [46]. Other studies postulated the formation of selenomethionine Se-oxide followed by deaminated selenoxide and further oxidized to Se(IV) species such as methylseleninic acid [48]. Dehydro-selenomethionine-oxide was also found in the presence of SeMet [49], while according to Larsen et al., SeMetO and methyloselenone are the possible reaction products [50].

The concentrations of MeSeCys and SeCys_2_, the oxidized form of selenocysteine, in their binary mixtures with catechin, were fast decreasing during the storage of their solutions (Figure 1). Particularly very rapid decline was observed for MeSeCys as its concentration decreased from 5.00 mg L^−1^ up to 0.16 mg L^−1^ after 2 days of mixture storage. No other selenium species were determined in that solution. According to literature data, the degradation of methylselenocysteine leads to the formation of methylseleninic acid and dimethyl diselenide [42,51]. It was indicated that SeMetCys degradation readily takes place in water extracts, but greater stability was observed in HCl and ammonium acetate solutions stored at low temperatures [52]. The instability of SeCys_2_ in different extraction media has been reported [52,53,54,55]. SeCys_2_ in a water extract from Se-yeast was relatively stable, particularly when the extracts were stored in a fridge or freezer [52].

The changes in the concentrations of epicatechin and Se(IV) or Se(VI) vs. prolonged time of their mixing, presented in Figure 2, are very similar to those determined for catechin mixed with these selenium species. In the mixtures of epicatechin with SeMet, the concentration of this flavanol decreased faster than in the case of catechin, while in the presence of SeMetO, the level of epicatechin stays almost constant. However, a rapid decrease in SeMetO was observed in the studied time range, and selenomethionine appeared after 4 h (about 1 mg L^−1^), which later was decreasing. In the mixtures of epicatechin and SeMet, the traces of SeMet were just noticeable after 8 h. Epicatechin with selenocysine disappeared also very fast as previously described in the presence of catechin and SeMetO was detected. After 36 h, its concentration was 2.1 mg L^−1^. The degradation of MeSeCys was not so fast as in the presence of catechin in the studied mixture.

EGCG did not change its concentration in the binary mixtures with Se(IV), SeMet, and MeSeCys (up to 24 h), but gradually decreased in the presence of Se(IV) and particularly SeMetO (Figure 3). SeMetO, SeCys_2,_ and MeSeCys showed the least stability in their binary mixtures with EGCG. In the case of selenocystine solutions, SeMetO was also detected in the concentration range of 0.65–1.45 mg L^−1^.

Catechins can also undergo degradation, oxidation, epimerization, and polymerization during food processing, and the stability of each catechin varies in different food matrices [12,49,56]. Many factors could contribute to their chemical changes such as temperature, pH, oxygen availability, the presence of metal ions, and the added ingredients. They are more stable in acidic media, but the addition of ascorbic acid was found to significantly increase catechin stability, whereas the addition of citric acid only had a minor effect [49]. EGCG, among tea catechins, is more susceptible to degradation during storage of the tea leaves. Vuong et al. reported that catechins may be degraded to form phloroglucinol carboxylic acid and protocatechuic acid [12]. In our study, the catechin solutions stored for 36 h at dark at room temperature exhibited good stability.

Figure 4 presents the changes in the concentration of selenium species in their standard solutions stored under the same conditions as the binary mixtures with catechin. The stability of Se species after 6 h of preparation decreases in the order: Se(VI)~MeSeCys > Se(IV) > SeMet > SeCys_2_ > SeMetO, while after 24 h and later this order looks like this: Se(VI) > SeMet > Se(IV) > MeSeCys > SeMetO~SeCys_2_. Thus, selenocystine appears to be the most unstable selenium species, similar to the mixture with catechin. It should be added that in the selenocystine standard solution after 16 h storage, the peaks characteristic for SeMetO and Se(IV) start to appear. Their concentrations were increased from 0.36 mg L^−1^ to 1.32 mg L^−1^ and from 0.10 mg L^−1^ to 0.79 mg L^−1^, respectively. Koterbai et al. reported that selenate, selenite, and methylseleninic acid could be the products of the oxidation of some selenoaminoacids [54]. However, in the case of mixtures of SeCys with catechin, Se(IV) formation was not observed.

The aqueous extract of different plants containing several compounds, such as flavonoids, alkaloids, saponins, carbohydrates, proteins, tannins, and steroids, is often used for the green synthesis of selenium nanoparticles [26,27,28]. Selenium precursor (Se(IV) compounds are the most often used) is mixed with the plant extract solution at different ratios and stirred for different periods, sometimes with heating. During this process, the color of the reaction media changes from colorless through orange, and brick red to red, indicating the formation of Se nanoparticles. Also, recorded UV–Vis absorption spectra are useful for the indication of the Se nanoparticles formation and can provide valuable information regarding their size [57]. According to this method, the suspension of SeNPs with ~20 nm diameter exhibits an absorption maximum of around 250 nm. For bigger nanoparticles, the characteristic red-shift of the absorbance peak maxima is observed as the particle sizes increase. SeNPs suspension with diameters of about 100 nm were characterized by an absorption maximum below 350 nm, while for SeNPs with diameters of about 250 nm, the absorbance maximum was observed at 680 nm.

The UV–Vis spectra of the binary mixtures containing Se(IV) and studied catechins recorded at various time intervals are present in Figure 5. It can be seen that with increasing mixing time, the location of the maximum absorbance peak was shifted a little toward longer wavelengths. That suggests that SeNPs changed their dimensions, probably by aggregation. The exception is the solution containing epicatechin and Se(IV), where this relation is reversed. Moreover, only for this mixture, a decrease in signal intensity was observed. Despite these differences, it can be supposed that the suspension of SeNPs with a diameter in the range of 100–120 nm is formed. For the mixtures of catechins with other selenium species, these characteristic peaks were not observed under studied conditions.

### 2.2. Catechins and Selenium Species in Tea Infusions

As tea aqueous extracts are a rich source of catechins, we examined the changes in the concentration of studied selenium species added (5.0 mg L^−1^) to black and green tea infusions. It was checked that these teas did not contain selenium. The obtained results are presented in Figure 6.

In the presence of both used tea infusions, the concentrations of selenium species decreased. This is particularly visible in green tea, where all selenium species, except selenites, gradually decreased during a prolonged time of their mixing. The concentration of Se(IV) was reduced much slower. The observed behavior of SeMetO, SeCys_2_, and MeSeCys is similar to that observed for their standard solutions alone (Figure 4).

SeMet concentration was faster reduced in the presence of green tea infusion, while in black tea, its concentration was almost stable in the interval time of 4–36 h. After 4 h for black tea and 8 h for green tea, the presence of Se(IV) was also recorded at its concentration equal to 0.80 mg L^−1^. While Se(IV) concentration with green tea infusion was almost constant during a prolonged time of storage, in the case of black tea infusion gradually decreased up to 0.16 mg L^−1^. Luo et al. found that microbes are involved in the oxidation of SeMet to Se(IV) in liquid media and the oxidation rate is more rapid than that of elemental selenium [58]. Earlier papers also reported that selenite can be one of the SeMet degradation products [46,54].

The composition of green and black tea infusions differ from each other in terms of polyphenolic content as well as the presence of other substances [29]. Therefore, the observed differences in the changes in selenium species concentration in tea infusions. The monomeric catechins and their derivatives are the major polyphenols in green tea, while black tea receives substantial oxidation, which results in the polymerization of catechins into theaflavins and thearubigins.

The recorded UV–Vis spectra of green and black tea infusions after the addition of Se(IV) or SeMet indicate the formation of SeNPs (Figure 7). Synthesized SeNPs were so strongly stabilized by the extract components that it was impossible to separate them from the solution by centrifugation. Thus, the spectra were recorded using a tea infusion as a blank. The reduction in selenite in the presence of black tea infusion is very fast, and the final product is received just after the reagents mixing. According to the method proposed by Lin [57], they should be assumed to have dimensions of about 100 nm or less. The prolonged time of reaction does not change the size of the obtained nanoparticles, which points to their good stability. The formation of SeNPs using green tea extract was much slower but with higher efficiency as the intensity of the recorded signals were twice as large. They exhibited the maximum at about 380 nm, which corresponds to their diameter larger than 100 nm. Green tea infusion, except for a high content of catechins, contains other compounds capable of reducing Se(IV), such as flavonols or ascorbic acid. The content of ascorbic acid is higher in green tea as it is less processed than the black type [29,59].

The mixtures of black and green infusion with added selenomethionine exhibited a maximum below 300 nm, indicating that the formed SeNPs have a diameter smaller than 100 nm. This maximum decreased over time, much faster in the solution with green tea.

In order to accurately determine the shape and size of the nanoparticles, they were subjected to DLS and TEM analysis (Figure 8). TEM analysis revealed that the nanoparticles are spherical in all cases. Moreover, previously obtained results regarding the size of SeNPs were confirmed and clarified. Those from black tea have about 89.0 ± 3.2 nm, and those from green tea have a size of 101.5 ± 4.2 nm. It can, therefore, be concluded that the nanoparticles obtained in black tea are capable of crossing the cell membrane and thus can potentially be used in medicine. The results obtained by the DLS method suggested that the SeNPs obtained are about 100 nm. Additionally, the polydispersity index (PDI), which is a measure of the heterogeneity of a sample based on size, was significantly higher for the SeNPs obtained in green tea. This means that the particles are differentiated in terms of size. In the case of SeNPs obtained in black tea, the PDI value was 0.109. According to the theory, if the coefficient is less than 0.100, the particle fraction can be considered homogeneous. Certainly, the SeNPs obtained in black tea are significantly more homogeneous than those from green tea.

It should be mentioned that the size of nanoparticles in biomedical applications has a strong effect on their interactions with living cells, influencing uptake efficiency, internalization pathway selection, and intracellular localization [60]. However, smaller NPs have a higher probability to crossing cell membranes and being internalized by passive uptake than large ones.

## 3. Materials and Methods

### 3.1. Reagents

All of the commercial standards of selenium species used in the experiments (Na_2_SeO_3_ (Se(IV)), Na_2_SeO_4_ (Se(VI)), selenomethionine (SeMet), methylselenocysteine (MeSeCys), and selenocystine (SeCys_2_)) were purchased from Sigma-Merck (Steinheim, Germany). Selenomethionine oxide (SeMetO) was synthesized based on the previously described method [43]. Briefly, 1.00 mL of 30% (*v*/*v*) H_2_O_2_ was added to 10.0 mL of an acidic solution of selenomethionine (10.00 mg L ^−1^).

The standards of polyphenolic compounds such as catechin, epicatechin, and epigallocatechin gallate, as well as the standards of green tea and black tea, were purchased also from Sigma-Merck (Steinheim, Germany).

For mobile-phase preparation, methanol hyper grade for LC-MS LiChrosolv was purchased from Merck (Darmstadt, Germany), and water was obtained from a Mili-Q water purification system (Millipore, Bedford, MA, USA).

### 3.2. Instrumentation

Quantitative and qualitative analysis of selected speciation forms of selenium and catechins was carried out using high-performance liquid chromatography in the HILIC mode. The used equipment was the Shimadzu LC system consisting of binary pumps LC20-AD, degasser DGU-20A5, column oven CTO-20AC, autosampler SIL-20AC, and 8030 triple quadrupole Mass Spectrometer (Shimadzu, Japan). The ionisation step was done using an ESI source operated in negative-ion (inorganic Se and polyphenolic compounds) or positive-ion mode (organic selenium species). The ESI conditions were as follows: the capillary voltage was 4.5 kV, the temperature was 400 °C, the source gas flow was 3 L/min, drying gas flow 10 L/min. Compounds were separated on Atlantis HILIC (100 × 2.1, 3 μm) from Waters (Dublin, Ireland). The mobile phase consisted of methanol and water (85/15, *v*/*v*) and was delivered at 0.2 mL/min. Selenium compounds were identified by comparing their retention times and m/z values obtained by MS and MS2 with the mass spectra. In addition, the mass spectrum of the entire sample was recorded to identify possible degradation products of selenium compounds.

The UV–Vis absorption spectra in the range of 250–900 nm were recorded using a Parkin Elmer (Waltham, MA, USA) Lambda 20 UV–VIS spectrophotometer. Data were processed with WinLab software version 2.85.04.

Selenium nanoparticles (SeNPs) were characterized using dynamic light scattering (DLS). The measurement was done on Mastersizer 2000 (Malvern Panalytical, UK) equipped with the wet samples dispersion unit (Hydro 2000 MU). The instrumentation was controlled using Malvern SOP operation software.

The size and shape of obtained SeNPs were examined using transmission electron microscopy (TEM) with a TALOS F200 model (Thermo Fisher Scientific, Waltham, MA, USA) working at an accelerating voltage of 200 kV. The preparation of the sample for measurement included spotting the post-reaction mixture containing SeNPs on a copper grid and then air dried before the examination.

## 4. Conclusions

The biofortification of plants with selenium compounds has already been established for creating Se-enriched agricultural products and overcoming human deficiency in this element. Plants take up inorganic selenium forms from the soil, which are then converted into organic species. On the other side, catechins are widely distributed in a variety of plant foods and also exhibit a beneficial effect on human health, mainly due to their antioxidant activity. Thus, this work investigated the interactions between plant catechins present in tea infusion and selenium species. The results from the conducted experiments were based on changes in the concentration of both reagents, their stability in aqueous solutions during a prolonged time, and the possibilities of SeNPs formation.

Single catechin solutions were stable under the experimental conditions, the same as selenate, while the content of remaining Se chemical forms gradually decreased. Selenium species exhibited also instability in the presence of studied catechins, and selenocystine appeared as the most unstable. The recorded UV–Vis absorption spectra confirmed the formation of selenium nanoparticles in the binary mixtures of catechins and Se(IV) only.

The formation of SeNPs was also observed when, except Se(IV), selenomethionine was added to black or green tea infusions, although in the mixtures of this selenoaminoacid with single catechin no characteristic peaks were observed. Partial degradation of SeMet to Se(IV) followed by its reduction to elemental Se may be an explanation for it. The formed nanoparticles were more stable when black tea infusion was used. From the point of view of SeNPs formation, it can be concluded that selenite as the most stable selenium species in the presence of catechins, could be used for the enrichment of teas providing good amounts when ingested.

The methods for the synthesis of SeNPs that involve naturally occurring substances in plant extracts are preferred as they could act both as reducing agents and stabilizers [26,27,28]. That approach requires non-toxic solvents, mild temperatures, and a reducing agent that is easily accessible, cheap, and not harmful to the environment. However, difficulties in their purification are a potential problem when their separation from the reaction solution is necessary. They were so strongly stabilized by the tea sample matrix that they did not precipitate out of the solution, even when centrifugation at high speed was used. However, for selenium supplementation, such suspension can be used, especially since SeNPs showed a greater ability to neutralize hydroxyl radicals compared to tea extracts [29].

On the other side, the bioavailability of the mixtures containing Se(IV) and tea catechins should be taken into consideration in health recommendations. The therapeutic potential of catechins is limited by their low absorption, poor pharmacokinetics, and bioavailability [61]. The metabolism process of SeNPs in the human body was very rarely studied [19,62], although they are considered as a novel Se supplements [63]. In addition, studies from experimental animals and epidemiological surveys have shown that both selenium species [14] and tea extracts [64] have dose-dependent toxicology. More selenium is not necessarily better for the prevention or reduction in disease, but its baseline status should be taken into consideration to avoid over-supplementation. Further studies will be undertaken to establish whether catechin and selenium species interactions alter their bioavailability.

## Figures and Tables

**Figure 1 molecules-28-05897-f001:**
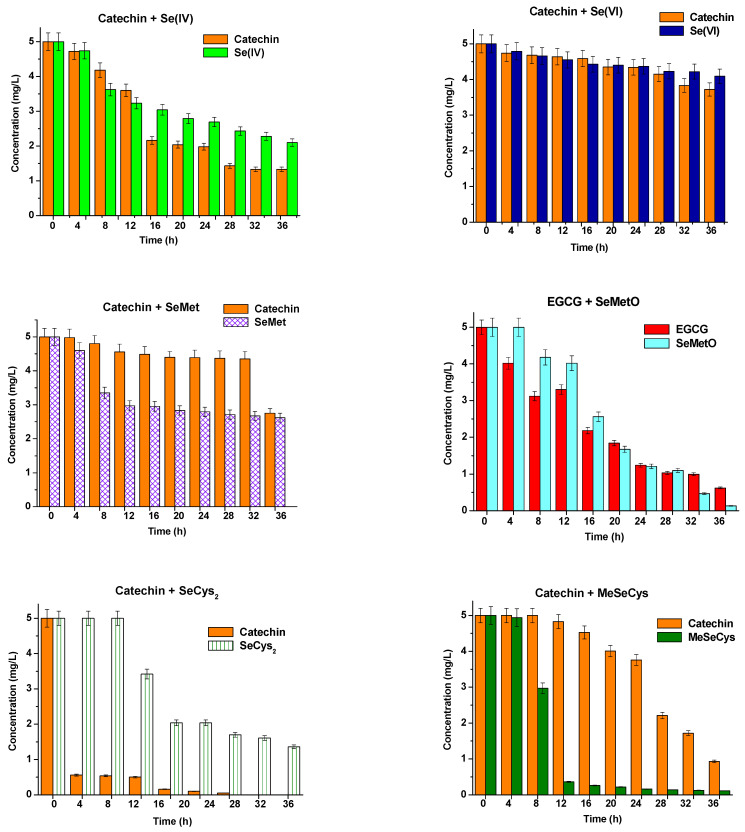
The changes in the concentrations of catechin and selenium species vs. time of their mixing. The results are expressed as a mean ± sd, n = 3.

**Figure 2 molecules-28-05897-f002:**
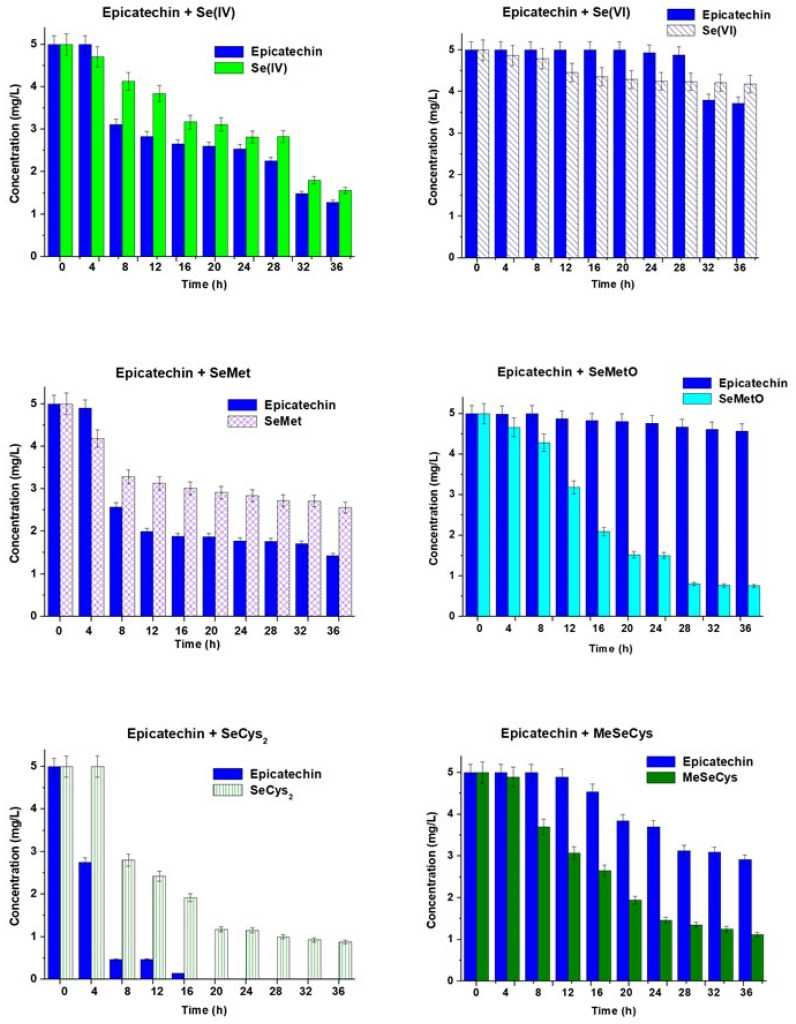
The changes in the concentrations of epicatechin and selenium species vs. time of their mixing. The results are expressed as a mean ± sd, n = 3.

**Figure 3 molecules-28-05897-f003:**
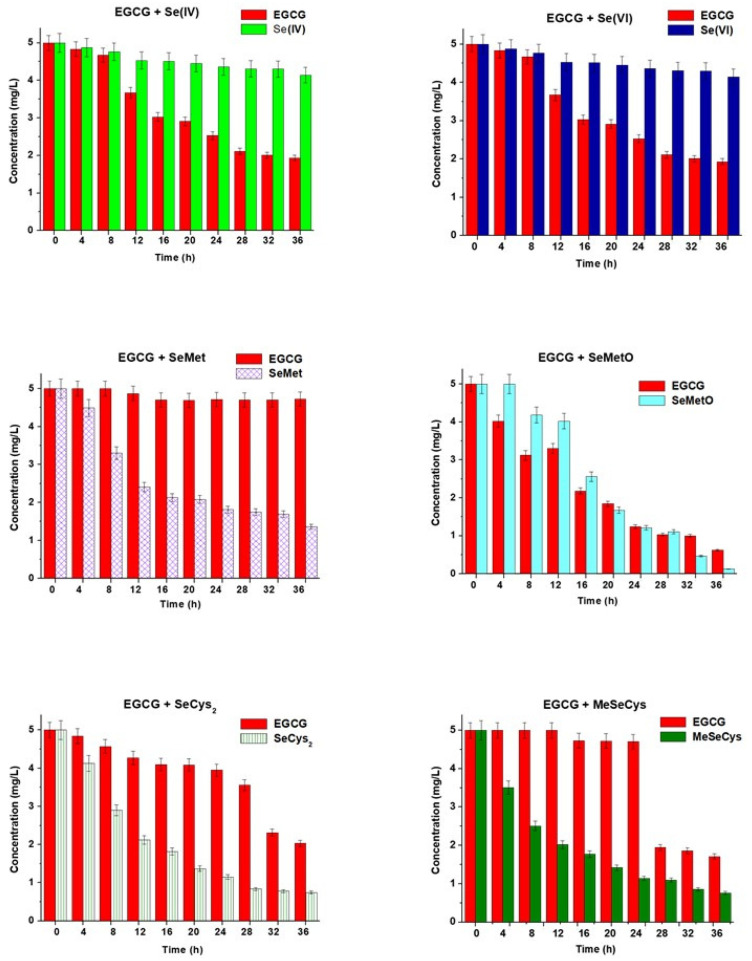
The changes in the concentrations of EGCG and selenium species vs. time of their mixing. The results are expressed as a mean ± sd, n = 3.

**Figure 4 molecules-28-05897-f004:**
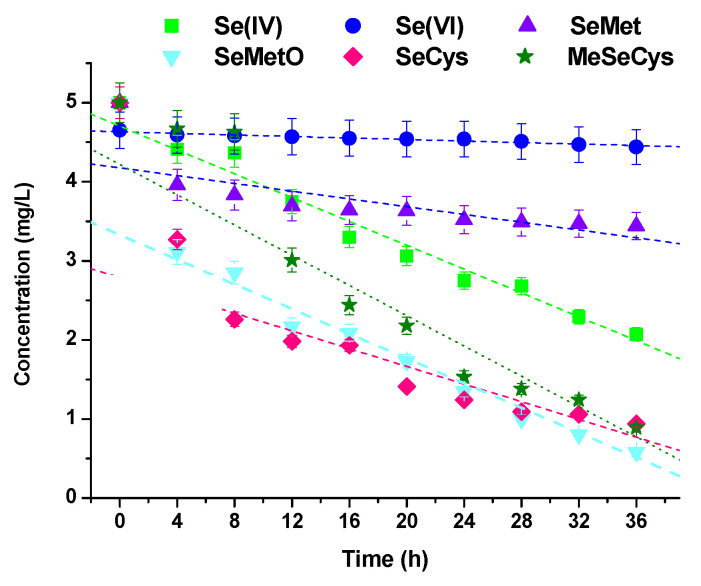
The changes in the concentrations of selenium species in their standard solutions vs. time.

**Figure 5 molecules-28-05897-f005:**
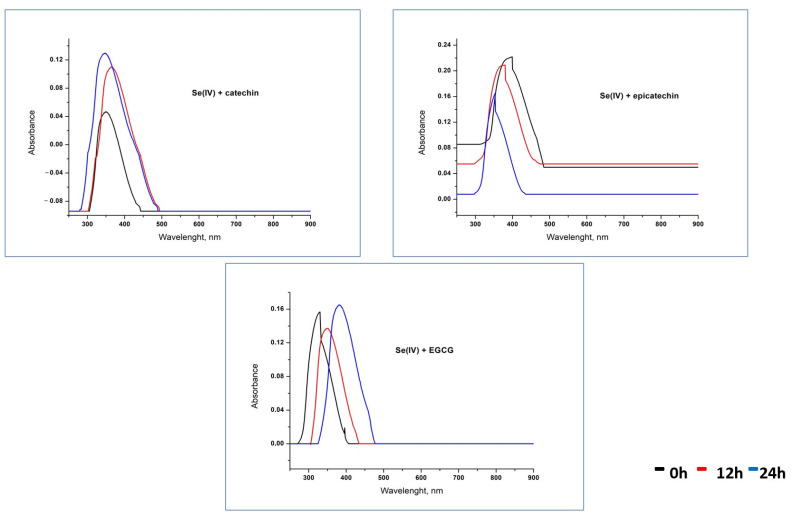
The UV–Vis spectra of Se(IV) and catechin mixtures.

**Figure 6 molecules-28-05897-f006:**
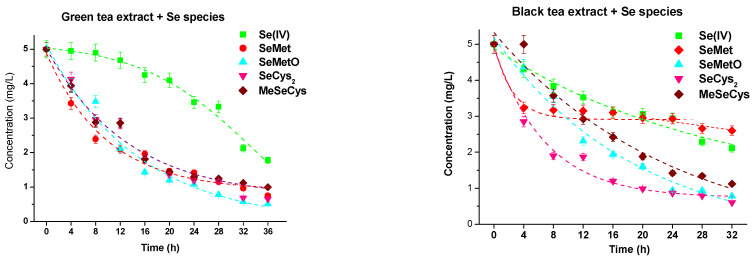
Changes in the concentration of selenium species added to green and black tea infusions.

**Figure 7 molecules-28-05897-f007:**
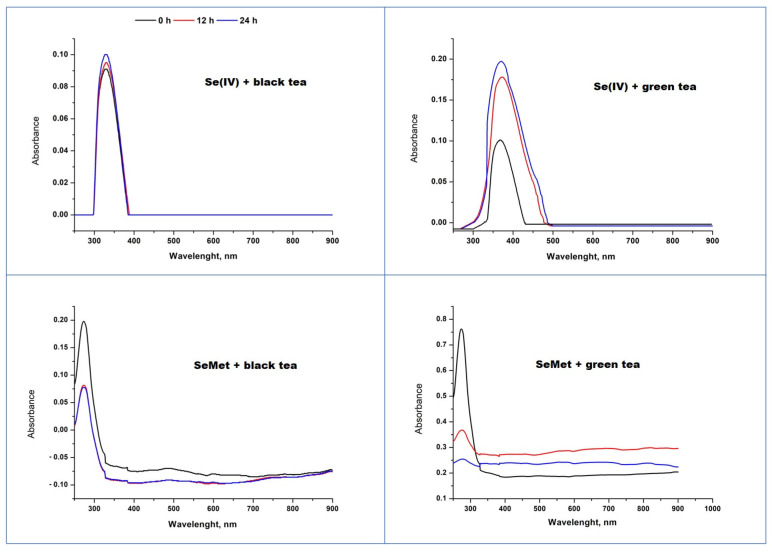
The UV–Vis spectra of green and black tea infusions with the addition of Se(IV) after prolonged times.

**Figure 8 molecules-28-05897-f008:**
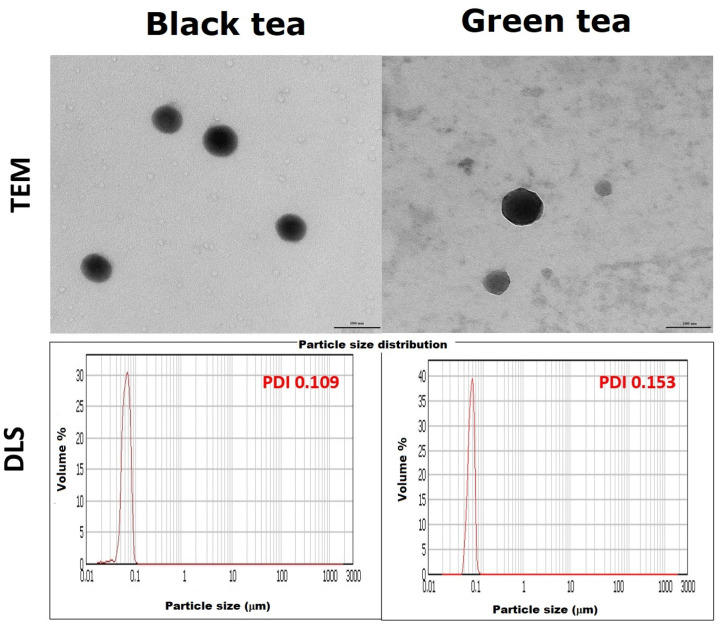
TEM images of obtained SeNPs and particle size distributions from DLS method.

## Data Availability

The data presented in this study are available on request from the corresponding author.

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
