# Peer review of "Catechins and Selenium Species—How They React with Each Other"

_molecules, 2023, doi:10.3390/molecules28155897_

Round 1

Reviewer 1 Report

The Nutraceuticals have great potential in promoting health. Many trace elements and phytonutrients are investigated for bioavailability in active form and for their stability, which are affected by exposure to enzymes, variations in pH and interactions with food matrix  when they are ingested. In addition ,two or more nutraceuticals may interact with each other affecting their bioavailability and stability. In the present manuscript, the authors have addressed these issues by using a model system consisting of flavonoids like catechins and trace elements like selenium in different forms. Selenium deficiency can lead to many health problems , which can be addressed by fortification in appropriate foods in desirable amounts.

The catechins and selenium play an important role in antioxidant defense systems and may control oxidative stress. The interaction between catechins and selenium may influence their ability to provide protection against oxidative stress . The work reported in this paper involves following interaction between different forms of selenium and catechins under simulated conditions. Further , these studies were replicated in tea infusions , which are naturally rich in catechins .The results indicated tat the catechin and selenium on their own are stable under experimental conditions. However, the chemical forms of selenium in the presence of catechins are unstable. Appropriate strategies need to be pursued to overcome instability of the two vital compounds to ensure that they are available in good amounts when ingested. 

Suggestions:

It is not clear from the study about the forms of selenium , which are ideal for fortification. Authors should make specific recommendations to overcome the pitfalls of adverse interaction between two selected nutraceutical molecules.

Presentation can be improved for clarity

Reviewer 2 Report

The article deals with the formation of nanoparticles. Nanoparticles are not shown either by DSL, TEM. How have SeNPs been characterized by size? 1) The abstract needs to be rewritten. Deciphering Se(IV) is required. It should be clearly shown what new results are obtained in this study. 2) In the introduction, the physical methods for obtaining selenium nanoparticles should also be discussed. Therapeutic Potential and Main Methods of Obtaining Selenium Nanoparticles - PubMed (nih.gov) The pros and cons of each method should be briefly described. 3) The obtained results are summarized in Figures 1-3. What is summarized in the figure? The data is not described. 4) The legends for the figures are not informative. How many repetitions were performed, what statistical methods were used and whether they were used at all.

5) Figure 5, 6.7. The figure quality is not acceptable. It is impossible to evaluate the results presented in the figure. 6) Discussion of the results should be presented as a separate chapter

Many parts of the text are difficult to read and require serious revision with the participation of a native English speaker.

Round 2

Reviewer 2 Report

The article does not introduce anything new in the field of research.